# Give Me Five: The Most Important Social Values for Well-Being at Work

**Reinaldo Sousa Santos** [1,*] and **Eva Petiz Lousã** [1,2]

1  Research Unit in Business Sciences and Sustainability (UNICES), University of Maia, 4475-690 Maia, Portugal
2  Centre for Organizational and Social Studies of Polytechnic of Porto (CEOS.PP), Polytechnic of Porto, 4200-465 Porto, Portugal
*  Correspondence: rsantos@umaia.pt

**Abstract:** Social values are very important for well-being at work. This study investigates which and how social values affect well-being at work and contributes to the growing interest that the issue of quality of life at work has aroused in the areas of human resources management (HRM). Semi-structured interviews were held with 21 active employees of a large Portuguese business group in the environmental sector. The study took place in two parts; first, in December 2018 and then two years after the start of the COVID-19 pandemic, in January 2022. Theories and concepts emerged from the thematic analysis and the subsequent consideration of the literature and emerging conceptual understanding. This qualitative interview study examines what employees expect from work experience about the behavior of leaders and supervisors as representatives of the formal structure of the organization and the behavior of co-workers as an expression of an ethical and positive work environment. The findings show the five social values most important for employee well-being: respect, trust, equity with no discrimination, help and gratitude. The knowledge of the social values with more impact on employee well-being constitutes very important information for human resource management and for the employees, themselves.

**Keywords:** human resources; employee well-being; social values; social interaction; respect; trust; equity and non-discrimination; helping and gratitude

## 1. Introduction

The 21st century is marked by two crises that have affected the organizational context: the financial crisis at the end of the first decade and, most recently, the COVID-19 pandemic. The financial crisis has shown the need to ensure the strengthening of ethical frameworks in the organizational sphere (Segon and Booth 2015), in line with the strengthening of social responsibility of organizations through the adoption of ethical and sustainable behaviors that respond to current needs without compromising the future (Minton et al. 2012). The COVID-19 pandemic had a strong impact on people's lives and, of course, also on work (Zhong et al. 2021), generating an unprecedented increase in working from home (WFH) to guarantee social distancing for employees (Moens et al. 2021; Schmitt et al. 2021). The spatial, instrumental, and relational configuration of the working day has changed profoundly, impacting on the way new and current employees live and build day-to-day working relationships and their general well-being (Stempel and Siestrup 2022; Zhong et al. 2021).

The growing attention given to ethics in the organizational context is part of a management paradigm that considers non-material purposes for organizations (Dickson et al. 2001), valuing the relationship with people, the community, and the adoption of citizenship behaviors (Hamel 2009). The processes associated with human resource management are highly sensitive to et-hics in organizations, as they have direct effects on people and are operated by them in the organizational context (Greenwood 2013). In contexts of crisis and uncertainty, ethical behavior increases the predictability of behavior generally and reinforces positive relationships and well-being at work (Colbert et al. 2016; Ragins and Dutton 2007).

The promotion of well-being at work, in addition to being associated with several positive results for people and businesses, has become an ethical duty of organizations (Guest 2017).

In face of this increasing attention to well-being at work (Guest 2017; Van De Voorde et al. 2012), several references to the importance of social values in the formulation of the social dimension of well-being emerge, especially in terms of trust and cooperation (De Clercq et al. 2019; Grant et al. 2007; Jena et al. 2018; Kulik 2021). Employees want to see congruence between their own social values and the social values that emanate from the organization's performance (Valentine and Barnett 2003). However, the literature's attention to studies of social values in the organizational context is still insufficient (Bolat and Korkmaz 2021; Christensen et al. 2021; Hemingway 2005; Jaakson 2010; Machaczka and Stopa 2022; Shafer et al. 2007), and it is important to identify which social values are most associated with well-being at work and to provide the human resource management with relevant information to reinforce the effectiveness of their intervention at work (Guest 2017). This information becomes even more important when work practices become more flexible and adjusted to the expectations of each employee, and the HRM focus is preferably directed towards the consistent production of positive impacts on the well-being of employees. The dimension "Employee Champion", associated with the mandate of HRM (Ulrich 1998), assumes an increasing importance in the organizational context with greater flexibility in activities and social relationships associated with work, demanding from organizations a consistent ethical performance, capable of obtaining approval in the scrutiny carried out by employees.

The present study focuses on addressing the need to understand in more depth which and how social values affect well-being at work. The study proposes to answer the following research questions: What are the social values that can most influence well-being at work? What was the impact of the COVID-19 pandemic on this association? The study intends to contribute to identifying the knowledge about the social values associated with well-being at work, and to add to the existing literature on WFH associated with the COVID-19 pandemic by looking at how an abrupt change has influenced employees' perceptions of their social values for well-being. Greater knowledge about the social values associated with well-being at work will be of great importance to strengthening the alignment and efficacy of HRM. The temporal proximity of the COVID-19 pandemic makes the contribution of this study to theory and practice very relevant.

In the following sections, we discuss the concepts of well-being at work, social values, and working from home. We then focus on the research approach, participants, interview protocol and data analysis. The results are analyzed and discussed in relation to the theoretical framework and consider the periods before and after the COVID-19 pandemic. We decided to merge the Results presentation with the Discussion to allow for a more detailed narrative, with greater attention to the detail of the participants' statements. However, in the whole section and for each theme there was a concern to start with the presentation of the results and only then move on to the bibliographic confrontation. Finally, the limitations, theoretical and practical implications and conclusions of the study are presented.

## 2. Well-Being at Work

Well-being can be conceptualized as a general quality of life or as a domain-specific concept (e.g., work-related well-being) (Taris and Schaufeli 2015). Well-being at work, also referred as "employee well-being", refers to the overall quality of an employee's experience and functioning at work. It includes affective, cognitive, and behavioral aspects associated with work (Guest 2017; Kazemi 2017; Kowalski and Loretto 2017; Wijngaards et al. 2021).

Well-being at work is a dynamic and multidimensional construct that encompasses psychological, physical, and social dimensions (Grant et al. 2007). Psychological well-being refers to the subjective experience associated with work and includes positive emotions (hedonic dimension) (e.g., Brown et al. 2012) and the fulfilment, meaning and purpose obtained at work (eudaimonia dimension) (Huta and Waterman 2014). Physical well-

being is associated with risk of injuries and illnesses to which the employee is exposed at work (Danna and Griffin 1999; Peccei et al. 2013), with a safe work environment having a positive impact on well-being at work (Guest 2017). It also includes the psychological and moral consequences associated with flexible and precarious work practices, intensification of the workload and the phenomena of moral and sexual harassment in the workplace (Ilies et al. 2010; Schneider and Harknett 2019). Social well-being, also called relational well-being (Guerci et al. 2019), designates the quality of interactions and interpersonal relationships in the work context (Grant et al. 2007; Guest 2017; Khoreva and Wechtler 2018; Peccei et al. 2013; Van De Voorde et al. 2012), illuminating issues such as trust and work-family conciliation (Edgar et al. 2017), fair treatment (Guerci et al. 2019), affective team commitment (Luu 2020), reciprocity, coordination, cooperation and integration (Grant et al. 2007). Social well-being includes the integration, acceptance, actualization, and coherence of social behaviors that occur at work (Colenberg et al. 2021; Kazemi 2017), making ethical scrutiny a very important dimension of well-being at work (Ehnert et al. 2016).

Well-being at work, although having received much attention in research in recent decades, remains a very superficial concept (Daniels et al. 2017) and relationships at work are largely ignored (Fisher 2010). However, evidence has been collected of the positive impact of the quality of social interactions at work for well-being, namely, to reduce conflict, stress, absenteeism, turnover, and health costs, as well as to increase employee satisfaction, resource sharing, engagement, and organizational commitment (e.g., Boreham et al. 2016; Colbert et al. 2016; DeGroot et al. 2000; Dimotakis et al. 2011; Feeney and Collins 2014). Employees with fewer sources of social support are at greater risk of suffering from health problems (Hammig 2017). Positive social relationships also reinforce instrumental and emotional performance, personal and career development, and constitute opportunities for friendships, with all the associated affective value (Colbert et al. 2016).

Considering the positive effects associated with well-being at work, human resource management began to pay increasing attention to its various dimensions (Boon et al. 2019). The promotion of well-being at work, with positive impacts on results and employees, has become an ethical duty of organizations (Guest 2017).

## 3. Social Values in the Organizational Context

The literature defines social values in the organizational context as a collective set of norms, values, and beliefs that express people's views in the workplace (Collins and Smith 2006) and guide how they evaluate, decide, and act on other people and situations (Schwartz 1999). All organizations have social values, although some are more results-oriented and others are more instrumental (Schein 2004). In the context of the organization, values signal what is right or wrong, and desirable or undesirable (Zwetsloot et al. 2013). Social values result from the perceptions that employees gather from the behaviors they observe in the work context (Walsh et al. 2018) and are fundamental to consolidate the sense of belonging to the group (Winter and Jackson 2014), as well as formalizing the psychological contract with the organization (Rousseau 2001). Values communicated and not practiced by the organization are highly destructive for organizational functioning and are associated with several negative outcomes such as employee dissatisfaction, leadership disbelief or loss of customers (Lencioni 2002). The alignment of employees with the real social values of the organization reinforces the intrinsic motivation and performance levels of employees (Beer 2009). The lack of coherence associated with social values strongly penalizes well-being at work (Kazemi 2017). Social values are transmitted to new colleagues through reception and socialization processes, and, in the face of behaviors that reinforce them, they will tend to persist in the future (Zwetsloot et al. 2013). Social values are, therefore, important instruments of human resource management, with a high capacity to affect performance and well-being at work, mainly through the quality, consistency, and coherence they give to social interactions that occur at work (Mowles 2008). The literature has identified some social values with relevance at work, for example interconnectedness, participation, trust, justice, responsibility, development and growth, resilience, respect, and competent and

responsible behavior (e.g., Strickland and Vaughan 2008; Zwetsloot et al. 2013). However, there is still a lack of research that, based on the perceptions reported by employees, contributes to illuminate the social values with more impact on the well-being at work and to strengthen HRM policies and practices.

## 4. WFH—Working from Home

The COVID-19 pandemic generated new trends in human resource management, with an accelerated reinforcement of work flexibility mechanisms, technological intermediation of processes, and internationalization and sustainability of operations (Cooke et al. 2021; Minbaeva 2020). The WFH increased abruptly and significantly (Kitagawa et al. 2021), stemming from an external context that went beyond the voluntary decision of organizations and people (Stempel and Siestrup 2022), and placing people in WFH who had never worked in this modality (Schmitt et al. 2021).

The impact of WFH associated with the COVID-19 pandemic on well-being at work is identified as an emerging and very critical issue for human resource management (Zhong et al. 2021), but the results are still contradictory and deserve further exploration. While more than 85% of employees report that they have lost well-being in life and work with WFH (Campbell and Gavett 2021), most employees in WFH want to continue in this modality, even after the end of COVID-19 restrictions (Felstead and Reuschke 2020). If WFH may have brought greater flexibility, autonomy and creativity to employees (Haufe 2020), on the other hand, it may have reinforced the risk of being always connected and needing to carry out digital detox actions (Schmitt et al. 2021; Vorderer et al. 2017); there are also the loss of social relationships at work and the increased risk of loneliness in life (Campbell and Gavett 2021). The study of social values associated with well-being at work gained an additional purpose in the face of the arrival of the COVID-19 pandemic, because it allowed us to compare the views before and during the use of the WFH.

## 5. Materials and Methods

### 5.1. Research Approach

A qualitative study was conducted to identify which and how social values affect well-being at work and the impact of WFH on social values for workplace well-being. Considering the exploratory nature of the study, the involvement of employees, and the desire to understand the phenomenon more deeply, it became necessary to move forward with a qualitative study (Edmondson and McManus 2007), through which it is possible to understand the processes of social construction that support the way employees build and understand their experiences in the organizational context (Gioia et al. 2013). The objective of the study does not go through the mere enumeration of social values associated with well-being, but it also intends to understand how this influence occurs, in the period before and after the pandemic.

### 5.2. Participants, Interview Protocol and Procedures

Participants are part of the largest Portuguese State Business Sector group in the environmental sector, which plays an important role in the fields of water supply and wastewater treatment. This group has exclusively public capital, although its employees have a private employment relationship. The organizations have between 93 and 363 employees and annual revenue between 14 and 88 million EUR. According to the Portuguese classification, 50% are considered medium-sized companies and 50% are large companies. The human resources teams of all group companies were contacted to participate in this study, but only four stated their interest in participating. The selection of respondents was the responsibility of each company, according to criteria of convenience and availability of employees/organization, being requested to represent different hierarchical levels, occupational areas, age, or gender. Twenty-one employees participated in the study. The group of participants included 43% men and 57% women, of which 52% were younger than 39 years old, 43% were between 40 and 49 years old and 5% were over 50 years old.

Regarding the function, 24% were operational technicians, 29% administrative technicians, 33% technicians with higher qualifications and 14% were team leaders. A total of 43% work in the Lisbon and Tagus Valley region, 29% in the south and 24% in the center of Portugal. The study took place in two phases: first in December 2018, and then two years after the start of the COVID-19 pandemic, in January 2022.

Data collection was carried out using semi-structured interviews to obtain more in-depth information about social values for well-being at work. The interview protocol was structured around the employee's social well-being at work, and the questions invited participants to share positive and negative situations at work and to describe the associated effects on their individual well-being. The topics of leadership, positive environment, and human resource management practices were also addressed. Initially the interviews were conducted in the workplace of each company, and all interviews were audio recorded via a portable computer device and had an average duration of 70 min. In the second phase, the interviews were carried out through electronic platforms (Microsoft TEAMS/Zoom) and had an average duration of 42 min. For both phases, each participant signed a consent statement authorizing the recording and use of the interview content for the stated purpose, as well as safeguarding their right to anonymity and confidentiality.

### 5.3. Data Analysis

The data analysis was carried out by using thematic analysis, which allows identifying, analyzing, and presenting patterns in the collected data. This is a method widely used within qualitative methodology to contribute to the understanding of explicit and implicit meanings associated with certain data expressed in text (Braun and Clarke 2006). We used inductive thematic analysis, as the content analysis of the interviews did not include deductive or pre-defined categories. The matrix of codes and themes resulted from the participants' contributions, evaluated in terms of frequency and positive or negative impact on well-being at work. The inductive approach is the most frequently used in the application of thematic analysis (Mills et al. 2010).

The content of each interview was transcribed verbatim and, before beginning the content coding phase, the data familiarization phase was advanced through an immersion exercise that "involves" repeated reading (Braun and Clarke 2006). The texts in the interview transcripts that appeared relevant were then highlighted and coded based on phrases used by respondents. The themes and subthemes were refined and revised with on-going data collection and fieldwork (Strauss and Corbin 1998). The 21 interviews proved to be enough, because the matrix of themes and subthemes was saturated by the sixth interview. To improve the reliability of the analysis, the categories and quotations were evaluated separately by two researchers.

### 6. Results and Discussion

The application of inductive thematic analysis allowed the identification of themes and codes expressed in the following Table 1, which also indicates the statistics of the associated registration units that characterize the relevant contributions of the participants:

**Table 1.** The Most Important Social Values for Well-Being at Work.

| Themes | Codes | Registration Units by Code (%) | Registration Units by Theme (%) |
|---|---|---|---|
| Respect: refers to friendly and dignified treatment | attention given to the employee as a person | 29% | 11% |
| | friendly behavior and language | 53% | |
| | appropriation of other people's work | 6% | |
| | attention to the employee's personal life | 12% | |
| Trust: refers to predictability of the other's behavior | response to uncertainty and insecurity | 30 | 18% |
| | emotional security | 10% | |
| | integrity and consistency | 23% | |
| | trust in supervisor | 23% | |
| | positive relationships | 14% | |
| Equity and non Discrimination: refers to fair equality between people | similar rights and duties as all others | 26% | 19% |
| | possibility of scrutiny | 16% | |
| | appreciation of merit and career progression | 19% | |
| | salary equity | 23% | |
| | sense of justice | 16% | |
| Help: refers to the voluntary action of helping another | personal duty | 27% | 40% |
| | common project | 39% | |
| | sense of autonomy | 8% | |
| | avoid an uncomfortable reaction | 5% | |
| | less attention to own work | 3% | |
| | positive impact on the lives of others | 18% | |
| Gratitude: refers to the positive recognition of what we have received | positive reciprocity | 60% | 12% |
| | feeling of belonging | 15% | |
| | response to voluntary help | 10% | |
| | response to external uncertainty | 15% | |

Considering well-being at work as a dependent variable in our study, these are the social values with the greatest capacity to influence it: Respect, Trust, Equity and Non-Discrimination, Help, and Gratitude. Next, we will present the results that justify the identification of each social value, and we will proceed to the discussion of these results in conjunction with the relevant literature. The conceptual map designed considers well-being at work as a dependent variable and strongly influenced by social values at work. Ethical and social validation of behaviors at work will generate more workplace well-being and, as a result, deliver more pleasure and fulfillment to people and more performance and sustainability to organizations.

### 6.1. Respect

Eight participants mentioned the importance of respect for well-being at work and shared several real situations of disrespectful and uncivil conduct that caused them revolt and suffering and severely penalized their well-being. Respect is one of the most basic and fundamental values for employees at work, as illustrated in the quote:

> *They made life black for us at all levels. Almost psychological torture. ( . . . ) So, I am telling you that receiving respect for me is everything.* (Rita)

The desire for respect in working relationships is common to all areas and functional levels. Participants mentioned that respect is manifested in the attention given to the

employee as a person and in horizontal or hierarchical relationships within the workplace. Respect essentially arises from attention to others, friendly treatment, and a lack of aggression. In line with the literature that regards respect as a positive consideration for the other person (Faulkner and Laschinger 2008), this value allows one to feel valued in the relationship (Laschinger 2004), reinforcing his or her self-esteem and well-being (Clarke and Mahadi 2017). Being treated with respect in the workplace means being treated with dignity as well as having the ability to make contributions for improvement and to be heard, valued, and recognized (Coetzee 2015). Respect disincentives rude or aggressive behaviors (Walsh et al. 2012).

Respect is also manifested in the employee's work experience and in the appropriation that he or she makes of his or her work, which we could call "owner of the work", as evidenced in the present study:

> *Also have respect for the work of others. I don't like bosses who take advantage of their employees' work. Our boss asks us for things, but then he doesn't have to brag about it, either, that he did it.* (Vera)

Respect for "work owner" fits into the notion of professional respect (Liden and Maslyn 1998), which establishes mutual respect for people's abilities (Graen and Uhl-Bien 1995).

During the WFH, the borders between workspace and family or personal space have blurred and interpenetrated more, and the respect manifested in the attention given to the personal life of the employee and in the working hours or schedule, as the participants pointed out:

> *People have gained more respect for their personal lives. Still, on Friday, it was five minutes to six o'clock, and a colleague called and immediately began to say, "I apologize for calling at this time." People try, as much as possible, to consider the separation of schedules for themselves and others.* (Vera)

As workers in WFH rely heavily on regular electronic communication, there was a greater concern about people choosing the communication channel to communicate with their colleagues and superiors, to avoid misinterpretations of what is communicated, as evidenced in the following statement:

> *For some people, it's better to call, because if we write, we need to put a lot of smiles because they can quickly get offended.* (Rita)

In turn, written communication allowed a reduction in the impulsive response in conflict situations:

> *(…) we have more time to think about the answer and how we should adjust it for each person. I got an email that freaks me out. In the office, I couldn't hide this reaction. Here at home, I get up, ( . . . ), think about the answer, and then answer more calmly. At work, with so many people on our side, we don't have this space to breathe and think. So, I think telecommuting allows you to reduce conflict.* (Rita)

However, some respondents highlight the preference for face-to-face communication when it is a more serious issue or a more delicate topic:

> *(…) to be more transparent and so that people can look you in the eye. Talking to the face shows respect for the other person.* (Queirós)

Face-to-face or videoconferencing interactions allow the exchange of non-verbal information between participants, which facilitates communication and avoids conflict (Schmitt et al. 2021). On the other hand, text-based communication is more prone to divergent interpretations and possibilities of conflict (Mahajan and Zaveri 2020; Riordan and Kreuz 2010).

*6.2. Trust*

Consistent with Grant et al. (2007), trust emerged as one of the key concepts for determining social well-being at work. Before the pandemic, trust was mentioned by half of the participants ($n = 12$), and after the pandemic, all participants mentioned that having rela-

tionships of trust became fundamental for their well-being at work. The distance associated with WFH has created insecurities that can be offset with more trust at work.

Relationships of trust make the self-behavior and other behavior predictable, in a logic of reciprocity that tends to endure (Robinson 1996). In line with Dunn et al. (2012), participants at the present study report that trust is associated with feelings of emotional security and with beliefs about the integrity and consistency of the other's performance. Trust stems from a significant commitment by those involved to the relationship of trust (Jena et al. 2018) and generates an affective bond between those involved (Colquitt et al. 2007). It also arises because of accumulated knowledge about the other person and an incentive to honor and not violate the trust that has been deposited (Kramer 1999). The commitment to non-violation of trust generates loyalty as a by-product of trust:

> *For me it is unthinkable to disappoint anyone who bets on me, who trusts me. For me loyalty is fundamental in any kind of relationship.* (Noémia)

Participants considered trusting relationships with leadership as very important for sharing the social value of trust in the organization. Trust of the supervisor materializes aspects of autonomy and recognition that the employees value, and translates into a feeling of security, as suggested by the participant Luís:

> *It's a security. He trusts the work I do, trusts the decisions, trusts what I'm doing. I feel a confidence, a security.* (Luís)

Supervisors convey the message that situations of poorer job follow-up, while possibly being associated with unavailability, constitute in themselves a posture of trust in the employee's autonomous performance. Indeed, trust in leadership is one of the important dimensions of ethical leadership, through which the leader acts and establishes interpersonal relationships that promote the dissemination of appropriate conduct values to subordinates (Brown et al. 2005; Kaishoven et al. 2011). Trust in management stems from the belief that management's own interests take no precedence over the employee's own interests (Ruppel and Harrington 2000). This performance can follow personal values associated with the attributes or motivations of the leader, or management values when translated into careful acting towards the subordinate, with impact on his or her professional or personal life (Chughtai et al. 2015). Participants also mentioned strengthening the trust relationship with the boss by reporting situations that showed attention to the employee's personal life:

> *For example, I have to go to a doctor appointment with my father and she said to me, "Go at ease". She knows, she trusts.* (Eduardo)

Participants noted the need for trust as an important factor in building positive relationships with one's co-workers, managers, and supervisors. The employee who receives the "vote of trust" is the one who feels supported and valued, as shown in the following excerpt:

> *I've had job offers to leave the company. No. I like being here, I feel supported, and I have a job near home. I like stability! And maybe a new challenge meant a lot of change in life. I'm right here. I have hope and trust in this.* (Belmiro)

The psychological contract is based on a relationship of trust between employee and employer (Rousseau 1989). Employees with high levels of trust in an organization are more likely to show commitment to the organization. Events that break the psychological contract destroy trust (Chiaburu and Byrne 2009).

Participants reported that WFH had a negative impact on trusting relationships at work, because physical distancing creates uncertainty about the behavior of others and many of the actions that created a sense of close relationship between people disappeared:

> *We end up having to validate if what they tell us happened through conversations with other people. If several people tell us the same version, then it must have happened anyway and then we come to trust.* (Susana)

> *My supervisor sometimes stopped by my office and shared with me some information that was not yet public. Even to know what I thought. I noticed that he trusted me. Now online these things no longer happen.* (Vera)

If organizations fail to balance the negative effects of physical distancing by enhancing internal communication and opportunities for interaction among employees, WFH can become a strong threat to trust and well-being at work and penalize stability and retention of work teams, as Paula warns us:

> *The lack of meetings, the lack of communication at the level of the team and the middle management, generated a climate that I can call mistrust. That's what made me want to change teams and roles, because I didn't feel like I was part of the team, I felt like I was here working alone. There was a lack of sharing, belonging, trust.*

### 6.3. Equity and Non-Discrimination

For half of the respondents (*n* = 12), the value of equity and non-discrimination is manifested in the fulfilment of the full rights of employees at work; the importance of this, as with trust, became more pronounced in the post-pandemic period. The increase in WFH and the greater difficulty in monitoring and scrutinizing the organization's decisions creates doubts about the possibility of not having access to rewards or opportunities offered to other colleagues.

The employee intends to enjoy similar rights and duties as all others who share his or her work experience, including opportunities for access to career advancement and participation in organizational events. The fundamental premise of non-discrimination and equal opportunities is based on the notion of fair equality between all persons, who, given similar effort and competence, should have similar prospects for success regardless of family social status, nationality, age, gender, religion, or any other factor not relevant for determining merit (Rawls 2001). However, the criteria of merit and uniformity do not always underlie access to certain opportunities but rather personal and family proximity, thus breaking the expectations of employees for equal opportunities as the report of one participant in the study suggests:

> *The message the company gives, and everyone knows, everyone talks about it is that they are friends and family and acquaintances. And I'm comfortable with this. This is public. So, when we understand this dynamic, we can't think "If I work hard, if I study, if I spend blank nights, I'll make it. No."* (Mafalda)

Such actions have a negative impact on the social well-being of employees and are considered morally unacceptable in the professional context, because they feel that their organization does not share the values it should. The desire for non-discrimination results from the individual and moral right of each person to consider himself or herself equal to others, and any differences in treatment should result from observable and morally recognized justifications (Arneson 2018).

Salary equity is another employee's expectation, whose compliance is determined comparing realities and situations and is operationalized through the perceptions that employees gather and not from consolidated facts and validated information (Shantz et al. 2018). In this sense, the perception of a good reward will contribute to employee social well-being. Participants are strongly penalized in their well-being with evidence of wage policy that does not comply with principles of equity as expressed in the following excerpt:

> *One came in, new people came in, with much less experience than us and a higher salary. We took the salary receipt and talked to her, and we know she [the director] went to the administration to talk to them.* (Catarina)

The value of equity and non-discrimination has several intersections with the value of justice. Sensitivity to justice is a fundamental characteristic of people (Wijn and Bos 2010), which justifies the natural tendency of employees to avoid proximity or involvement in situations of injustice (Hamlin 2014; Jensen et al. 2014).

When participants were in WFH, they showed greater difficulty in ethically scrutinizing the actions of the organization, management, and colleagues:

*Since we were away, there were some situations of attribution of benefits to colleagues that we did not know about.* (Vera)

Physical distance also promotes the increase in rumors and non-validated information, as well as impairing the perception of equal opportunities for career development and progression:

*If I spend a lot of time telecommuting, I think I might get overlooked in the face of career advancement opportunities. Looking into the eyes and intervening on the spot about what is happening is completely different than being at home. I have no doubt that people's careers will get worse in this regard. Either the performance appraisal changes, or I don't know how my boss can measure certain parameters when I'm at home.* (Queirós)

However, the negative effect of physical distancing can be mitigated when employees share trusting relationships:

*I trust my supervisor and that's why I'm calm and don't feel discriminated against. I do not think about it.* (Susana)

*6.4. Help*

Almost all participants (*n* = 18) expressed satisfaction and interest in sharing a work environment that highlights helping behaviors among employees. Interestingly, with the rise of WFH, employees' attention shifted to other social values, and help was reported less. As clarified below, the sense of autonomy associated with the WFH reduces the need for help.

A common feature of these statements is the notion of common project sharing, which makes it possible to reconcile wills so that the success of one will be the success of all, translated into expressions such as "sharing the same boat" and "wearing the shirt". The duty to provide support to others generally arises as the standard posture appropriate for the work context. While for some, it is an inevitability that stems from personal personality and identity, for others, it is a consequence that stems from the content of one's function, as a better way of working, and there is a desire for more time to support the others.

*I help because it's part of me. It's already part of me.* (Rita)

*Helping and being helped is important. We all need support when we don't know. I think we should have more availability for that.* (Filomena)

Organizational literature has devoted attention to helping others at work (De Clercq et al. 2019). Voluntary action to help others stems from an individual ethical drive (Turnipseed 2002). Providing help is assumed to be an important social value that conditions employees' relationship actions with others in the work context (Deckop et al. 2003; Turnipseed 2002), contributes to improving the carer's performance, organizational effectiveness, and employee well-being (De Clercq et al. 2019), as well as improving the overall organization (Sparrowe et al. 2006; Van Dyne and LePine 1998). Helping behavior is one of the integral parts of organizational citizenship behavior (Gabriel et al. 2018). However, the literature has been warning of a "dark side" of organizational citizenship behaviors (Bolino et al. 2013), with negative impact on well-being (Koopman et al. 2016), work-family reconciliation (Halbesleben et al. 2009), and career progression (Bergeron et al. 2013). Participants also noted some negative effect of helping behavior on doing their own work, as expressed in the following excerpt, but they consciously consider that the "bright side" of helping has a higher value than the "dark side":

*Sometimes people call ten, fifteen, twenty times. They interrupt a little, sometimes we lose focus, we are working on other things. But I feel good to be able to help people. Of course, I do.* (Queirós)

An important feature that stands out from the participants' reports is the operationalization of the help, not only to help the needy colleague, but to avoid an uncomfortable

reaction in a confrontational scenario in the face of the denial of the help requested or visibly seen as necessary.

> *He was down there, and I was up here. I wasn't doing anything, and he comes in and he could say: "You didn't even go down there. At least you went down there to see what was going on." They don't get upset, but there's already a needle there, as I say. The more needles we stick in, the worse it is. When we help there are no fights, no hassles, and the thing flows, we are a team, and we move forward.* (Octavio)

This statement suggests how a culture of mutual help, truly rooted in the organization and that makes collaborative behaviors predictable and recommended, has the capacity to drive individual helping behavior at work. According to social exchange theory, voluntary actions are motivated by the expected and predictable return (Blau 1964). Participants report that help takes place in reciprocal cycles, which ensure that each employee has the support they need. According to the reciprocity norm, employees will tend to adopt workplace help behaviors in order to make predictable the possibility of also receiving workplace help (Deckop et al. 2003).

The satisfaction of helping others is in line with the benefits and roles attributed to positive relationships at work (Colbert et al. 2016; Podlewska 2016). One of the few moments when one of the participants cried during the interview was when she recalled a past work experience in which she was responsible for evacuating the company's employees from a conflict zone abroad. The emotion came from the realization that the health and life of others was in her hands, and under very difficult and delicate conditions, she was able to respond and bring them all safely home:

> *I think it was the day I felt the most ... There are no words [cries]. Because I didn't leave anyone there. I don't know if you understand, one of the people came in a private charter.* (Rita)

WFH has turned employees away from their potential sources of help on the job. As a result, the WFH contributed both to increasing the employees' sense of autonomy, and reinforced the sense of abandonment and lack of protection of employees with less technical and social resources:

> *As we are far away, before asking for help, each one started to look for and solve it alone.* (Vera)

> *With distance, we don't know the difficulties the other person has. If she's next door, we'll be able to notice and help. From a distance, if she doesn't take the initiative to talk to us, we don't know. So, I think that from a distance, people can be more unprotected in case of difficulty.* (Susana)

*6.5. Gratitude*

Gratitude was a social value mentioned by about half of the participants (*n* = 12) as an important value for their well-being at work. While the other social values were sometimes illuminated from the sharing of situations of violation and discomfort, all references to gratitude emerged expressly associated with well-being. It should be noted that none of the team leaders mentioned it, which could mean that leadership is not attuned to the expression of gratitude, with a negative impact on the well-being of the leader and the team.

Gratitude arises in response to helping and positive relationship behaviors at work and feeds a cycle of positive reciprocity that tends to generate more help and more gratitude:

> *There is a certain gratitude and a desire to go back and give a little more to try to compensate for the attention they had with us.* (Amilcar)

Gratitude also appears as a positive feeling of belonging to a team or organization that provides services or products with importance or notoriety recognized in the community.

> *I like the work I do. (...) After all, it is gratifying to hear people say that they consume our products.* (Eduardo)

The expression of gratitude is assumed as a strong message of belonging and cohesion between members of the same team or organization. As Tomás says, expressing gratitude means being on our side: *We see when people say thanks, because we know when people are on our side.* The emotion of those who are pleased to help may not be different from the emotion of those who are pleased to be helped. The participants expressed high gratitude for the situation in which they benefited from the voluntary help of co-workers, as well as opportunities that the organization has provided them for personal or professional development. Gratitude includes feelings of appreciation and positive affection for those who provide help (McCullough et al. 2002). More than a conjunctural characteristic, the literature suggests that it is a relatively continuous and stable characteristic of the person to recognize the positive contribution of others (Watkins et al. 2003), a life orientation that enhances the positive aspects of the world (Wood et al. 2010). Positive psychology has paid attention to the phenomenon of gratitude (Peterson and Seligman 2004), identifying that the grateful person has greater satisfaction and well-being (Lyubomirsky et al. 2005; Emmons and McCullough 2003) and less likely to express depression or stress (Cho 2019). Gratitude at work, although a topic understudied, has high potential to positively influence employee well-being and organizational performance (Cain et al. 2019).

With the WFH, the participants reported a reinforcement of gratitude towards their working conditions, due to the perception of greater insecurity in the labor market and in society:

> When we see the news about people who are out of work, who have financial difficulties, we cannot complain about that, our salary remains the same every month, we still receive it, we still have this possibility of being able to stay at home in telework and to be able to manage our personal life. I feel gratitude for being in this situation, even because I know very complicated cases. (Susana)

Well-being at work, in contexts of greater insecurity, tends to approach Rabbi Schachtel's (1954, p. 37) sentence: "happiness is not having what you want, but wanting what you have."

*6.6. WFH and Social Values*

Our investigation shows the abrupt change to WFH associated with the COVID-19 pandemic has influenced employees' perceptions of their social values for well-being. In this context, respect for the personal and professional lives of employees, as well as for their working hours and schedules, has emerged as a social value that contributes positively to well-being at work. WFH can undermine trust and well-being at work and penalize stability and retention of work teams. Recent studies have reinforced the importance of organizations adopting consistent and transparent internal communication actions to strengthen relationships, reduce uncertainty, and create an atmosphere of safety, well-being, and motivation among employees in WFH (Kim et al. 2021; Li et al. 2021).The WFH has shown several threats to the sense of equity perceived by employees, namely in terms of protecting their privacy and penalizing the most fragile social and professional groups at work (Bonacini et al. 2021; Charbonneau and Doberstein 2020; Meyer et al. 2021; Williams and Kayaoglu 2020). Leadership must be committed to creating and strengthening an environment of equality at work (Lee 2020). The protection of a sense of justice is fundamental to well-being at work and differential treatment at work must be justified on socially accepted reasons (Arneson 2018). "Far from sight, far from the heart", is not an acceptable reason.

WFH is identified as a source of greater flexibility and autonomy at work and as an opportunity for jobcrafting with a positive impact on well-being at work (Azizi et al. 2021; Haufe 2020; Ipsen et al. 2021). However, strengthening autonomy requires technical and social resources that are not equitably distributed among people working from home. Helping behaviors in WFH will benefit from a positive environment that fosters cooperation and protects employees from the emotional exhaustion associated with isolated work or less social interaction (Stempel and Siestrup 2022). The present study suggests that in

the context of WFH, people who feel grateful at work are more likely to be satisfied with their personal and professional lives. Gratitude at work has high potential to positively influence employee well-being and organizational performance, according to a study by Cain et al. (2019).

## 7. Conclusions

The aim of our study was to investigate which and how social values affect well-being at work. We conducted semi-structured interviews with 21 participants before (2018) and after (2022) the COVID-19 pandemic. The content was analyzed according to thematic analysis, using an inductive approach, and the five social values with a strong impact on employee well-being were identified: Respect, Trust, Equity and Non-Discrimination, Help, and Gratitude. These findings allow us to corroborate the values of trust (Zwetsloot et al. 2013), respect (Strickland and Vaughan 2008), adding the importance of values of equity and non-discrimination, help, and gratitude to well-being at work. The enumeration of social values and the presentation of the impact of WFH on well-being at work responds, even if not completely, to the need to understand how the pandemic has changed work practices and affected the well-being of employees. This information is essential to support management decisions in organizations committed to promoting well-being at work. Greater understanding of the social values connected with workplace well-being is critical to improving HRM alignment and effectiveness. The social values of an organization are conveyed to new colleagues through the processes of reception and socialization, so in the face of reinforcing behaviors, they will tend to last over time (Zwetsloot et al. 2013). Social values are assumed as management tools (Mowles 2008) and although reported values are only the "tip of the iceberg" (Jaakson 2010), based on them is the idea that the employee acts reciprocally in the organization (Rogozińska-Pawełczyk 2000). The identification of social values associated with well-being at work is an important contribution to HRM, which is increasingly concerned with valuing well-being at work (Azizi et al. 2021; Hamouche 2021; Guest 2017; Kowalski and Loretto 2017) and acting ethically aligned with the principles of social responsibility (Castillo-Feito et al. 2022). HRM can transmit and solidify social values through its daily activities (Kulik 2021), thus strengthening the organization's corporate image and benefiting from all the positive effects that literature associates with well-being at work.

In theoretical terms, our results reinforce the importance of ethical and socially validated behavior in the social relationships that occur at work. Participants declare that their level of well-being is associated with the perception of the behaviors they observe at work. This permanent scrutiny involves observing the performance of co-workers and, even more prominently, the supervisor as a hierarchical representative of the organization. As a result, these findings highlight the need for human resource management to promote a positive work environment that allows positive relationships between employees, guided by friendly and collaborative behavior, which reduces uncertainty about the future and reinforces the sense of a common project. Efficient management of selection, compensation management, performance evaluation, training and development and career management become critical to guarantee non-discrimination of employees and to satisfy their legitimate expectations of career progression and salary increase through merit and good performance. Working at work or at home, the employee sends a "Give Me Five" to his or her employer. If he or she gets the five social values that are important to his or her well-being, he or she will be committed. If the answer is not positive, then the employee will feel a violation of his or her psychological contract with the organization and resign and leave (Muresanu 2017; Rousseau 1989). Or worse still, resign and stay.

## 8. Limitations and Future Directions

Our study has the innovative effect of reconciling the enumeration of social values associated with well-being at work with the assessment of the impact of WFH during the post-pandemic period. The research has given voice to employees belonging exclusively to

a Portuguese publicly owned business group, which could compromise the generalization of our results to other types of organizations and countries. It is relevant to assess in future research if this organizational idiosyncrasy has an impact on the inventory of fundamental social values. WFH is an unavoidable issue in HRM during and post-pandemic (Zhong et al. 2021). However, the specificity of the COVID-19 situation raises questions about the generalization of results beyond the pandemic. Future, post-pandemic studies in distance working modalities are needed to deepen how social values can be strengthened or weakened and impact the well-being of employees.

**Author Contributions:** Conceptualization, R.S.S. and E.P.L.; methodology, R.S.S. and E.P.L.; software, R.S.S. and E.P.L.; validation, R.S.S. and E.P.L.; formal analysis, R.S.S. and E.P.L.; investigation, R.S.S. and E.P.L.; resources, R.S.S. and E.P.L.; data curation, R.S.S. and E.P.L.; writing—original draft preparation, R.S.S. and E.P.L.; writing—review and editing, R.S.S. and E.P.L.; visualization, R.S.S. and E.P.L.; supervision, R.S.S. and E.P.L.; project administration, R.S.S. and E.P.L. All authors have read and agreed to the published version of the manuscript.

**Funding:** This research received no external funding.

**Institutional Review Board Statement:** The qualitative research was carried out in accordance with the principles of the Declaration of Helsinki and the obligation to comply with the requirements established in the GDPR—General Data Protection Regulation was expressed.

**Informed Consent Statement:** Informed consent was obtained from all subjects involved in the study.

**Data Availability Statement:** Data is available upon request and upon institutional approval.

**Conflicts of Interest:** The authors declare no conflict of interest.

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
