# Peer review of "Give Me Five: The Most Important Social Values for Well-Being at Work"

_admsci, doi:10.3390/admsci12030101_

Round 1
Reviewer 1 Report
First, I would like to congratulate the authors for their work and contribution to a very interesting topic. Nevertheless, I do consider that the suggestions below will add value to the paper:
-a detailed and more thoroughly developed methodology, indicating the advantages and limits of the method(s) used, the details of interviews, questions
-provide more comments on the qualitative results obtained and correlate with other research in the field, making more comparisons with previous studies or indicating alternative methods;
-why your methodology is better than others?
-what are the gaps covered and the benefits for the literature?
Author Response
All suggestions and improvements indicated have been accepted and allow us to improve the content of the article. Thank you very much. Attached is a revised version of the article, with revisions in track changes mode. Comments identify the reviewer and subject of the major revisions report. Best regards from the authors

Reviewer 2 Report
Please see the comments in the attachment.

Author Response
All suggestions and improvements indicated have been accepted and allow us to improve the content of the article. Thank you very much.
We were unable to access the changes indicated in the text of the article. Perhaps this document has not been uploaded.
Attached is a revised version of the article, with revisions in track changes mode. Comments identify the reviewer and subject of the major revisions report.
Best regards from the authors

Round 2
Reviewer 1 Report
The paper was much improved although there is enough place for new additions.
Author Response
Dear Reviewer
As suggested, we made several additions to the article:
- Introduction:
We have included more information about the research question and relevance of the study. At the end of the Introduction, a brief presentation of the structure of the article has been included.
- Method:
We have included more information on the thematic analysis approach used: inductive.
- Results:
A table identifying the themes and codes resulting from the thematic analysis was included. The table also indicates the frequency statistics associated with each code and theme according to the relevant contributions of the participants.
- Conclusions:
We have included a summary of the main characterization information of the study and results and we have withdrawn the discussion carried out on the WFH - Working from home. This component moved to Results and Discussion, in which a specific section was created. We also made the contribution of the study clearer in theoretical and practical terms.
For updates to the article, see the comment: Minor Revisions
Best regards from the authors.

Reviewer 2 Report
Since the authors did not receive my edits, I will not hold them responsible for the editorial changes suggested re the first draft.
Author Response
We appreciate all your contributions to improving the article. Thank you very much.